# Review of Osteoradionecrosis of the Jaw: Radiotherapy Modality, Technique, and Dose as Risk Factors

**DOI:** 10.3390/jcm12083025

**Published:** 2023-04-21

**Authors:** Erkan Topkan, Ahmet Kucuk, Efsun Somay, Busra Yilmaz, Berrin Pehlivan, Ugur Selek

**Affiliations:** 1Department of Radiation Oncology, Medical Faculty, Baskent University, Adana 01120, Turkey; 2Clinics of Radiation Oncology, Mersin City Education and Research Hospital, Mersin 33160, Turkey; 3Department of Oral and Maxillofacial Surgery, Faculty of Dentistry, Baskent University, Ankara 06490, Turkey; 4Department of Oral and Maxillofacial Radiology, Faculty of Dentistry, Baskent University, Ankara 06490, Turkey; 5Department of Radiation Oncology, Bahcesehir University, Istanbul 34349, Turkey; 6Department of Radiation Oncology, School of Medicine, Koc University, Istanbul 34450, Turkey; 7Department of Radiation Oncology, MD Anderson Cancer Center, The University of Texas, Houston, TX 77030, USA

**Keywords:** radiotherapy, osteoradionecrosis, radiotherapy modality, radiotherapy technique, dose-volume parameters, radiobiological basis

## Abstract

Radiotherapy (RT) or concurrent chemoradiotherapy (CCRT) is the cornerstone of organ-sparing or adjuvant therapy for nearly all head and neck cancers. Unfortunately, aggressive RT or CCRT can result in severe late toxicities, such as osteoradionecrosis of the jaws (ORNJ). The incidence of ORNJ is currently less than 5–6% due to advances in dental preventive care programs, RT planning systems, and RT techniques. Although numerous patient-, tumor-, and treatment-related factors may influence the incidence rates of ORNJ, RT modality (equipment), technique, and dose-volume-related factors are three of the most influential factors. This is mainly because different RT equipment and techniques have different levels of success at delivering the prescribed dose to the focal volume of the treatment while keeping the “organ at risk” safe. ORNJ risk is ultimately determined by mandibular dose, despite the RT technique and method being known predictors. Regardless of the photon delivery method, the radiobiological effects will be identical if the total dose, dose per fraction, and dose distribution within the tissue remain constant. Therefore, contemporary RT procedures mitigate this risk by reducing mandibular dosages rather than altering the ionizing radiation behavior in irradiated tissues. In light of the paucity of studies that have examined the impact of RT modality, technique, and dose-volume-related parameters, as well as their radiobiological bases, the present review aims to provide a comprehensive overview of the published literature on these specific issues to establish a common language among related disciplines and provide a more reliable comparison of research results.

## 1. Introduction

Head and neck cancers (HNC), which account for nearly 6% of all cancer cases worldwide, are the sixth most common cancer type [1]. HNC affects many subsites and necessitates multimodal oncologic therapy for treatment. Support groups for occupational therapy, smoking cessation programs, speech and swallowing therapy, physical therapy, oral and dental care, and nutritional needs are also essential. Radiotherapy (RT) plays a pivotal role in the oncological management of HNC. Radiotherapy is the only curative alternative for patients with early-stage squamous cell HNC who are medically inoperable. It is also the only option for inoperable locally advanced salivary gland carcinomas where effective chemotherapy alternatives are not accessible [2]. Locally advanced HNC (LA-HNC) patients may benefit from RT as an adjuvant therapeutic option after surgery and as the backbone of organ-sparing treatment when combined with concurrent chemotherapy [3]. Furthermore, RT can be an effective primary palliative strategy for recurrent or metastatic disease [4].

With the exception of laryngeal cancers, most HNCs manifest as locally advanced HNC (LA-HNC), in which case definitive concurrent chemoradiotherapy (CCRT), either alone or in combination with induction chemotherapy, represents the gold standard of care in terms of an organ-sparing approach [5]. Yet, in a substantial proportion of patients, aggressive RT or CCRT may cause severe late toxicities such as submucosal fibrosis, muscular stiffness and pain, restricted neck movements, dysphagia, xerostomia, tooth loss, trismus, and osteoradionecrosis of the jaws (ORNJ) [6,7]. ORNJ is one of the most feared complications of head and neck RT or CCRT. The prevalence of ORNJ varies from 0.4 to 56% depending on the year of publication, how the condition is defined, and the RT modalities used [8]. Due to advances in dental preventive care programs, RT planning systems, and RT techniques such as intensity-modulated RT (IMRT) and intensity-modulated proton therapy (IMPT), the incidence of ORNJ is currently less than 5–6% in the majority of publications [8,9,10,11,12]. Although the impact of improved dental care programs cannot be underestimated, this reduction in ORNJ incidence is primarily due to the highly conformal dose distribution provided by cutting-edge RT techniques, which enable strict restriction of high RT doses to target volumes while sparing neighboring healthy tissues [13,14]. Nevertheless, ORNJ may still affect a significant proportion of HNC patients depending on the tumor and nodal localization and bulk, as well as inescapable mandibular high RT doses due to the tumor’s proximity to the mandible [15,16].

The mandible is more susceptible to ORNJ development than the other facial bones, which may be due to the mandible’s lower vascular supply: only one-sixth of the maxilla [17,18,19,20,21]. Another possible explanation is that the jaw is more frequently enclosed in the radiation portal and receives higher RT doses than the maxilla, which may be a risk factor for increased ORNJ rates [22]. Many patient-, tumor-, mandibular surgery-, dental procedure-, and treatment-related risk factors have been linked to an increased risk of ORNJ development, with most still being debated [17,18,19,20,21,22,23,24,25,26,27]. The RT modality, technique, and the dose received by the mandible are three of the most significant risk factors for ORNJ, and they are inextricably linked. Although RT modality and technique are often used interchangeably, they are actually quite distinct; RT modality refers to the treatment machine capabilities and radiation type, whereas RT technique refers to the type of treatment planning. Despite the notable influence of each of these three parameters on the incidence rates of ORNJ, they are often undervalued in comparison to conventional risk factors. Therefore, the primary objective of this review is to provide a concise overview of the established and frequently contested risk factors associated with the RT modality, technique and the dose received by the mandible for ORNJ, along with a brief radiobiological basis for their role in this severely incapacitating RT complication.

## 2. Impact of Radiotherapy on Bone Turnover

### 2.1. Predisposing Anatomical and Physiological Characteristics

The mandibular processes gradually grow and fuse in the midline during the fourth and fifth weeks of the embryological period. The left and right Meckel cartilages, which are cartilaginous rods that serve as the cores around which the membranous bone of the lower jaw develops, are made by neural crest cells of the first pharyngeal arch between the fifth and eighth weeks of development. The regions of the lower lip, lower jaw, and lower cheek are formed by the mandibular processes. The mentum designates the location of the midline fusion of the two mandibular processes.

The mandible is a flat bone with two rami and a body that forms a horseshoe shape. Apart from the middle ear bones, the mandible is the only moveable bone in the skull and the largest bone in the viscerocranium, or fascial skeleton. In contrast to the other skull bones, which are joined by sutures, the mandible articulates with the surrounding bones via a synovial joint called the temporomandibular joint. This anatomical characteristic makes it possible to chew and speak while maintaining the mandible’s connection to the skull.

The mandible receives its main blood supply from the inferior alveolar artery, which also nourishes the nerves, gingivae, and teeth. It is a small muscular artery that arises from the first portion of the maxillary artery and runs alongside the inferior alveolar nerve, the main nerve that innervates the mandible and its associated structures. The inferior alveolar artery splits into the mylohyoid and lingual branches before to entering the mandibular foramen. It then gives rise to the incisor arteries as it traverses the mandibular foramen and the mental artery as it leaves the mental foramen. It is worth noting that the inferior alveolar artery can occasionally arise from the external carotid artery rather than the maxillary artery and can be found in duplicated form. Necrosis of the mandible and structures perfused by its branches is possible after an injury or occlusion of the inferior alveolar artery.

In healthy adults, the mandible, like other bones, is a dynamic tissue with a balance of bone formation and resorption—the remodeling process—that is regulated systematically by hormones and locally by growth factors and cytokines [28,29,30]. The mandible has a compact-to-trabecular bone ratio similar to the distal radius and other long bones: 80% compact and 20% trabecular bone [31]. Due to its unique architecture of trabeculae, plates, and rods, the mandible has ten times the surface area of compact bones, resulting in a more pronounced remodeling process owing to an abundance of endosteal surfaces and cells [32,33]. The rate at which trabecular bone in the mandible remodels is six times faster than in the femur and twice as fast as in the maxilla [34]. There is also a difference in remodeling rates between the parts of the mandible. The rate of alveolar bone turnover at the alveolar crest, for example, is twice as fast as that of the mandibular canal and three to five times faster than that of the inferior compact border of the mandible [35]. Regardless of the type of injury, any damage to the mandible may logically result in a faster devitalization of the bone tissue than its comparators, including the maxilla. This difference is partially due to the mandible’s restricted blood supply, which is nearly one-sixth that of the maxilla, making the mandible more susceptible to necrosis than the maxilla due to a delayed or insufficient repair process in the absence of sufficient oxygenation, where the supply cannot meet the increased demand.

### 2.2. Radiobiological Considerations

Recent decades have seen significant technological advancements that have improved the quality of localized RT. While delivering ionizing radiation to the tumor is critical, minimizing damage to surrounding healthy tissue remains a significant obstacle despite these advancements. Effective management of RT’s early and late adverse effects requires knowledge of how healthy tissues respond to ionizing radiation. Radiation exposure has been shown to cause adverse changes in the physiologic levels of many factors, including cytokines, which play a crucial role in the onset of radiation-induced normal tissue injury [36]. Ionizing radiation causes the immediate formation of highly reactive free radicals following exposure, culminating in immediate protein changes and damage to cell membranes, ribonucleic acid (RNA), and deoxyribonucleic acid (DNA). A chronic increase of reactive oxygen species (ROS) is created and maintained for long periods as a consequence of secondary reactions after the initial hydrolysis of water and direct ionization following radiation exposure. The production of ROS by various cell types during injury and inflammation can also result in reactive nitrogen species, which can further exacerbate oxidative damage [37]. Damaged endothelial, epithelial, and inflammatory cells, for instance, may intensify localized oxidative stress and long-lasting radiation injury by producing chemicals like superoxide and nitric oxide [38]. Superoxide is a very toxic NADPH metabolite that is produced as a result of radiation-induced injury, which may destroy any sort of cellular component, including DNA, either directly or by producing other hazardous secondary radical species [39,40]. The direct effects of radiation occur when it interacts directly with the atoms of the DNA molecule or another vital biological component. However, because the amount of DNA molecule is too small, the likelihood of radiation interacting with it is less probable. Therefore, it is more likely that ionizing radiation will exert the majority of its effects indirectly (almost two-thirds), by ionizing water molecules and producing free radicals that lead to double-strand DNA breaks. Regardless of the mechanism of action, these early wounds may cause changes in cellular vitality and functioning as well as a local or systemic inflammatory response that may last for months or years, depending on the inherent radiosensitivity of the injured cells [41].

Bones are highly specialized anatomical structures susceptible to the damaging effects of ionizing radiation, especially at higher dosages. The bone is a common location of radiation-induced damage because of its high calcium concentration and propensity to absorb roughly 40% more radiation than surrounding tissues [42]. Radiation causes an excessive release of cytokines and chemokines during the damage response, leading to the symptoms of acute inflammation such as increased vascular permeability, localized edema, endothelial cell death, and vascular thrombosis [42,43]. In the later stages, RT also promotes fibroatrophic processes, which cause tissue to have poor vascularization and obstruct effective repair. This adverse circumstance renders tissue more fragile and triggers inflammation to flare up or reoccur after local injuries, such as tooth extraction or dental implant placement procedures. Similar to osteoporotic diseases, RT decreases the trabecular bone volume and skeletal stem cell populations while raising bone marrow adiposity and serum CTX/TRAP5 levels, resulting in more sluggish and less effective fracture repair [44,45]. Blood levels of osteocalcin and TRAP5 quickly increase after bone irradiation, indicating an increase in osteoclast activity [46]. At 12 weeks post-irradiation, trabecular bone volume decreases significantly, likely due to drastically reduced osteoblastogenesis, whereas osteoclastogenesis returns to nearly normal levels [46]. Such observations confirm the emergence of a decreased bone formation-to-resorption ratio and deteriorated bone quality. Irradiated bones’ skeletal stem cells appear to favor adipogenesis over osteogenesis, which further contributes to RT-induced bone loss. The rapid increase in osteoclast activity after RT and the gradual decrease in osteoblast activity in the weeks that followed are thought to be responsible for this phenomenon [45,47]. High-dose RT may alter the differentiation characteristics of skeletal stem cells in favor of decreased differentiation potential but increased radiation-induced cellular senescence, as evidenced by a robust galactosidase labeling signal that overlaps with cell death patterns [44,48].

The hallmarks of ORNJ include the development of hypovascular, hypocellular, and hypoxic bone and soft tissues, along with a chronic inflammatory process that worsens with time. These RT-induced changes increase cell death and collagen breakdown above and beyond the standard homeostasis capacity of cell repair and collagen synthesis, resulting in fibroatrophic and necrotic bone formation [42,49]. In a ground-breaking study, radiation-free samples from HNC patients were compared to radiation-treated samples from 40 ORNJ patients who received 50.4–70.4 Gy [42]. According to a histopathology examination of the bone and soft tissue samples, hyperemia and endarteritis were the early effects of irradiation that persisted for up to 6 months after exposure. Irradiated bone specimens had more cellular loss than their soft tissue counterparts, and indications of increased hypocellularity emerged quickly after irradiation in bone. The vascular structures were found to contain dense fibrous material years after being exposed to radiation, providing proof that thrombosis had formed. It was also discovered that radiation-induced damage had end-stage indicators like a decline in vascular content and an increase in tissue fibrosis, both of which got worse over time [42,49].

Modern molecular research has demonstrated that damaged tissue may also have metabolic derangements, which may be attributed to hypoxia, inflammation, and the activation of various pathways, such as hypoxia-inducible factor-1 alpha (HIF-1α) and the mechanistic target of rapamycin [50]. Irradiation-induced HIF-1α enhanced signaling induces the overproduction of transforming growth factor β1 (TGF-β1) and vascular endothelial growth factor (VEGF) [51,52]. TGF-β1 is a multifunctional cytokine secreted latently by cells. However, radiation-induced ROS exposure and the actions of activated proteinases cause the active TGF-β1 protein to be released and bind to one of several TGF-β1 receptors. Immediately following irradiation, a rise in TGF-β1 expression may be observed, and the increase is dose and time-dependent [53,54]. Although the TGF-β1 expression may normalize after this initial spike, it often rises again in chronically injured tissue after radiation exposure [55]. Numerous experimental and clinical studies have proven that TGF-β1 plays a role in clinical radiation damage. TGF-β1 exerts its action through canonical (Smad-dependent) and non-canonical (Smad-independent) mechanisms. The canonical route is crucial to the regulation of fibrosis genes such as collagen I, collagen III, and fibronectin. Non-Smad TGF-β1 activated signaling pathways, including Rho/ROCK, p38, and JNK, seem essential for fibrosis regulation. Even though TGF-β is a growth factor that is known for being anti-inflammatory, the interaction between TGF-β-R1 and TGF-β-R2 can cause upregulation of TRAF4 and TRAF6, which in turn activates inflammatory mediators like the multifunctional transcription factor NFκB. The expression of COX-2, iNOS, and STAT3 can be induced by NFκB, which can exacerbate fibrosis, EMT, and inflammation. TGF-β1 is one of the most prevalent cytokines released after tissue exposure to radiation, and its release correlates directly with radiation dose [56], indicating that TGF-β1 is a crucial marker of radiation toxicity and DNA damage in irradiated cells. If a cell survives DNA damage, the first genetic consequence of ionizing radiation is the increased production of TGF-II [57]. Consequently, it is plausible to hypothesize that radiation-induced release of TGF-β1 may influence fibrotic cell death, the final stage of ORNJ development; however, additional fundamental research is required to confirm this hypothesis. But a few reliable studies, like Delanian’s research on the fibroatrophic theory, have shown that ROS and TGF-β1 play significant roles in the early inflammation, fibrosis, and remodeling that result in terminal tissue necrosis in the development of ORNJ [42]. Similar hypothetical findings were also reported by Lyons et al. [58] and Bras et al. [59], lending support to the role of radiation-induced cytokines, fibrosis, and vascular abnormalities in the pathogenesis of ORNJ.

A precise balance must be maintained between bone formation by osteoblasts (OBs) and bone resorption by osteoclasts (OCs) to maintain a healthy bone microenvironment and a functioning skeletal system throughout the course of a person’s lifetime [60]. Additionally, healthy levels of various hormones and cytokines are necessary for strictly and appropriately regulated bone metabolism; however, any dysregulation in this complex system can lead to osteoporosis or osteoporotic diseases depending on the dominant remodeling process unrelated to the bone type. Tumor necrosis factor-alpha (TNF-α) is a pro-inflammatory cytokine that plays a crucial role in maintaining bone homeostasis by inhibiting osteoblast activity and boosting osteoclastogenesis during bone remodeling. However, TNF-α also contributes to the development of chronic inflammation. In confirmation, TNF-α has been shown to play active roles in the development of inflammatory joint disorders such as rheumatoid arthritis (RA), which causes extensive juxta-articular bone degradation, and ankylosing spondylitis (AS), which causes simultaneous bone breakdown and excessive formation. Existing in vitro research has shown that TNF-α inhibits osteoblastic differentiation and stimulates osteoclastogenesis through differential expression of many transcription factors, including NF-κB. TNF-α together with interleukin-1 (IL-1) has been shown to induce osteocyte apoptosis. Because they attract osteoclasts, apoptotic osteocytes have a significant impact on osteoclastic bone resorption during bone remodeling, which is in part regulated by TNF-α. Furthermore, TNF-α has been shown to significantly reduce F-actin levels, nitric oxide (NO) production, and intracellular calcium. This physiological imbalance may result in a decrease in osteocyte elasticity, providing a potential mechanism to explain how inflammation contributes to bone mass loss. Briefly, TNF-α disturbs bone homeostasis by activating osteoclastic resorption, suppressing osteoblastic proliferation and matrix production, and activating TNF receptor-associated Factor-2 (TRAF-2), which stimulates NF-κB, AP-1, and MAPKs signaling pathways, resulting in reduced bone formation. In this regard, it has been noted that proinflammatory and proresorptive IL-1, IL-6, IL-17, and TNF-α are significantly elevated 24 to 48 h after radiation exposure, which causes early bone loss after irradiation, confirming the crucial roles of proresorptive and inflammatory cytokines in post-irradiation bone loss [61,62]. Considering that TNF-α is an inflammatory cytokine involved in the acute phase reaction and is rapidly and persistently expressed in irradiated and adjacent tissue, it is reasonable to assume that rapid rise of TNF-α after irradiation may also play significant roles in ORNJ formation after head and neck RT. But before any meaningful conclusions about the precise function of TNF and related cytokines in ORNJ development can be drawn, this inference needs to be examined in thoroughly conducted pathophysiologic research.

According to the findings of basic research that provided the basis for Marx and colleagues’ 3H (hypovascular, hypocellular, and hypoxic) theories and Delanian’s radiation-induced fibroatrophic ORNJ theories [48,54], hypoxia, inflammation, and related cytokines appear to play a significant role in ORNJ genesis. Microvascular damage and subsequent vascular occlusion following RT are additional drivers of compromised bone integrity, which leads to hypovascularity, hypocellularity, local hypoxia, and fibroatrophic healing. After irradiation of a well-designed minipig model, the endothelial linings of vascular structures edematized in just 1-day, followed by the obliteration of small luminal vessels, indicating an induced hypoxic condition [63]. Although there was a transitory increase in blood flow two weeks after irradiation, this was followed by a progressive decline, demonstrating that microvascular damage occurred far sooner than bone damage. It further confirms the centrality of vascular occlusion and the associated chronic hypoxia in the evolution of ORNJ, which may set in motion a self-perpetuating cycle of persistent inflammation and heightened fibrosis. Dekker et al. [64] recently provided new evidence that radiation exposure causes microvascular damage in human mandibles. The authors assessed 20 irradiated, edentulous patients who had received mandibular dental implants, with the radiation-free implant patients acting as the control group. Bone biopsies at doses >50 Gy showed reduced vascular density and preferential obliteration of microvascular structures in the irradiated group. Clinical evidence also supports a vascular origin for ORNJ given that it can occur up to six times more frequently in the mandible than in the maxilla, which has better blood flow [65,66,67]. In a recent clinical trial including 263 patients with locally advanced nasopharyngeal carcinoma (LA-NPC) who underwent CCRT, Yilmaz et al. [68] shown the relevance of hypoxia in the development of ORNJ. The connection between pretreatment hemoglobin (Hb) levels and ORNJ rates served as this study’s primary outcome measure. The authors reported an 8.7% ORNJ prevalence rate. The optimal pre-CCRT Hb cutoff was 10.6 g/dL. When patients were divided into two groups based on this criterion, the Hb ≤ 10.6 group had a considerably higher ORN rate (32.5% vs. 1.5% for Hb > 10.6; *p* < 0.001). The encouraging findings of this pivotal study may serve as the foundation for additional basic and clinical research examining the crucial role of systemic and local hypoxia in the genesis of ORNJ, even though Hb is only an indirect indicator of tissue hypoxia.

ORNJ is a multifaceted complication of RT in patients with HNC, involving multiple physiological disruptions and suppressed or activated cytokines. Despite the substantial body of literature associating tissue hypoxia, elevated apoptosis, chronic inflammation, and hyper-fibrosis with the emergence of ORNJ, no clinical study has been reported that evaluates the potential utility of related biomarkers in the precise prediction of HNC patients receiving RT or CCRT (Figure 1). The exception to this is the most recent work by Yilmaz and colleagues [68]. If these findings are replicated, it might pave the way for a better understanding of ORNJ’s complicated biology and the creation of new, more effective approaches for preventing and treating this devastating disease.

### 2.3. Radiotherapy Modality and Technique

For radiation-induced toxicity in general and ORNJ in particular, the RT modality (equipment) and technique are two of the most influential factors (Table 1). This is mainly because various RT techniques have varying degrees of success when delivering the prescribed dose to the treatment’s focal volume while still ensuring the safety of the “organ at risk” (OAR). Optimizing tumor coverage by focusing high doses on the affected area while limiting exposure to healthy tissue is a top priority in today’s RT techniques. In contrast to the primary tumor and lymphatic regions, which can receive nearly comparable doses with any 2-dimensional RT (2D-RT), 3-dimensional conformal RT (3D-CRT), IMRT, or heavy ion therapy, such as proton therapy and carbon ion therapy, doses to OARs can only be reduced to desired values using computer-aided, sophisticated treatment plans and their delivery by modern treatment machines (Figure 2).

If the mandible is designated as an OAR during treatment planning, the use of IMRT, image-guided radiation therapy (IGRT), or hadron therapy may reduce the doses received by the mandible and, consequently, the risk of ORNJ. Therefore, in addition to the other OARs, the mandible should be distinguished as a separate OAR, and mandibular dosages should be kept as low as possible to reduce the risk of ORNJ in HNC patients. Maesschalck et al. [69] compared the incidence of ORNJ after IMRT in comparison to 3D-CRT techniques. The cohorts included 145 patients in the 3D-CRT group and 89 patients in the IMRT group. Total incidence rate of ORNJ was similar for both groups with rates of 11% versus 10% (*p* = 1.0). Nevertheless, in contrast to the findings of Maesschalck and colleagues, existing research data often indicate that cutting-edge RT techniques result in reduced ORNJ incidence rates. Nguyen et al. [10] analyzed 83 patients treated with definitive CCRT, post-operative RT/CCRT, or RT alone to determine the efficacy of IMRT and IGRT in reducing the risk of ORNJ. The mean mandibular doses for IMRT and IGRT were 43.6 Gy and 43.8 Gy, respectively, with only 1 (1.2%) ORNJ incidence at a median follow-up of 28 months, which is less than the generally cited range of 2% to 28%. These results lend credence to the efficacy of advanced RT techniques in reducing the risk of ORNJ, even though the median follow-up time was shorter than the commonly mentioned 36 months for ORNJ development. The prevalence and risk factors for ORNJ in patients with OCC and OPC were investigated by Moon et al. [70]. Out of 252 consecutively treated patients, 14 (5.5%) were found to have ORNJ upon review of their medical records. On univariate analysis, factors associated with ORNJ included a primary diagnosis of OCC versus OPC [hazard ratio (HR): 3.0; *p* = 0.04], continued smoking during RT (HR: 3.1; *p* = 0.04), mandibular invasion of the primary (HR: 3.7; *p* = 0.04), tooth extraction before RT (HR: 4.52; *p* = 0.01), and treatment with 3D-CRT versus IMRT (HR: 5.1; *p* = 0.003). The presence of pre-RT dental extractions, as well as the RT technique, were both confirmed to be significant by multivariate analysis. Aarup-Kristensen and colleagues examined the prevalence of ORNJ and associated risk variables in 1224 HNC patients treated with 66–68 Gy [8]. IMRT was used to treat the vast majority of patients and controls (94%). Cases of ORNJ were identified through cross-referencing the national Danish Head and Neck Cancer database with clinical observations at follow-up and hospital code diagnostics following oral-maxillofacial surgery. Documentation of dental procedures, including mandibular surgery, performed prior to RT for patients with ORNJ cases and two controls (1:2) was collected in a nested case-control study. ORNJ was observed in 56 cases (4.6%), with a median time to occurrence of 10.9 months (range: 1.8–89.7) following RT; 90% of cases occurred within 37.4 months. This study’s authors hypothesized that IMRT was responsible for the significantly lower ORNJ incidence rate (4.6% vs. 21.0%) they observed compared to the previously published DAHANCA 7 trial, in which 3D-CRT was the primary RT technique. Recently, Yilmaz et al. [68] utilized IMRT in 263 locally advanced nasopharyngeal carcinoma patients and reported 8.7% ORNJ rate. A recent meta-analysis by Balermpas et al. [71] provided additional support for these studies by showing that fewer than 5% of patients who underwent tooth extractions before IMRT experienced ORNJ as a side effect.

Theoretically, compared to IMRT (photon RT), more advanced intensity-modulated proton therapy (IMPT) may further lower ORNJ rates due to its inherent physical properties. Protons stop in the tissue after depositing their maximum energy (Bragg Peak), leading to a reduced integral dose and sparing of normal tissue. A constant relative biological effectiveness of 1.1 relative to photon RT is used for treatment planning. However, it is worth to note that increased relative biological effectiveness values may occur particularly at the field edges, which are usually located in the normal tissue due to clinical safety margins. Zhang et al. [72] compared mandibular doses and ORNJ rates in patients with oropharyngeal cancer (OPC) after IMRT or IMPT. A total of 584 patients who received definitive RT were included. Mandibular doses (minimum: 0.8 vs. 7.3 Gy; mean: 25.6 vs. 41.2 Gy; *p* < 0.001) and ORNJ (2.0% vs. 7.7%; *p* = Not specified) rates were lower for patients treated with IMPT than that with IMRT. These findings showed that using IMPT decreased excess irradiation of the mandible and, as a consequence, the risk of ORNJ for OPC. Singh et al. [15] recently published their IMPT experience with 122 OPC patients at Memorial Sloan Kettering Cancer Center. During a median follow-up of 40.6 months, ORNJ was documented in 13 (10.6%) individuals. The most involved ORNJ location in this investigation was the posterior ipsilateral mandible inside the radiation field that received the entire targeted IMPT dosage. This disheartening ORNJ rate indicates that ORNJ continues to be a clinical challenge even in the era of highly conformal IMPT, despite its unquestionable dosimetric advantages.

Along with the dosimetric benefits of IMPT, carbon ion therapy, a different type of hadron therapy, also has radiobiological benefits. However, no significant research findings in this field have been released as of yet. A retrospective evaluation by Musha et al. [73] included 199 HNC patients who received carbon-ion therapy. However, only 11 individuals with OPC and floor of mouth cancers were examined. The treatment consisted of 57.6 Gy or 64 Gy (relative biological effectiveness) administered in 16 fractions. The association between radiation dose and mandibular ORNJ was investigated. ORNJ was diagnosed in 5 (45.5%) of the patients after a median follow-up of 68 months. With cut-off values of 16.5 cc (*p* = 0.002) and 1.8 cc (*p* = 0.0059), respectively, doses of 30 Gy (relative biological effectiveness) to the mandible and teeth had the most striking impact on ORNJ development.

Although the dose distribution characteristics of IMPT and carbon ion therapy may offer superior tissue sparing than photon-based RT techniques, data on ORNJ is limited, and the results are contradictory. For instance, compared to the frequently cited <5% ORNJ rates with IMRT, the recently reported 10.6% ORNJ rate with IMPT in the study by Singh et al. seems discouraging [68]. Therefore, it is essential to conduct large-scale, prospectively evaluated hadron therapy studies in order to reach more reliable conclusions regarding this issue of utmost importance.

### 2.4. Radiotherapy Dose

Even though RT modality and technique are established predictors of ORNJ, the dosage to the mandible is the ultimate determinant of the probability of this complication. If the total dosage, dose per fractionation, and dose distribution within the tissues remain the same, the radiobiological consequences will be identical regardless of the photon delivery technique. Consequently, modern RT techniques appear to mitigate this risk by reducing mandibular doses rather than by altering the behavior of ionizing radiation in irradiated tissues. Over the years, numerous studies have evaluated the risk of ORNJ in HNC patients based on different mandibular dosimetric parameters (Table 2). Despite significant methodological differences, the vast majority of studies have consistently suggested a strong correlation between mandibular dosages and the risk of ORNJ: the higher the dose, the greater the risk of ORNJ.

Kubota et al. [74] reviewed the medical records of 616 patients with HNC who were treated with definitive or postoperative RT to analyze dose-volume histogram (DVH) parameters related to ORNJ. Forty-six (7.5%) patients experienced ORNJ after a median follow-up period of 40 months (range: 3–145 months) and a median time to ORNJ of 27 months (range: 2–127 months). The mandibular body was the most frequent location for ORNJ (83%). After analyzing the DVH data, the researchers found that ORNJ patients’ V30-V70 values were significantly higher than those without ORNJ. Univariate analyses revealed that the primary tumor site, pre-RT mandibular surgery, post-RT tooth extraction, and V60 > 14% were all significant factors; however, only the tumor site (*p* = 0.0059) and V60 > 14% (*p* = 0.0065) remained significant in multivariate analyses. The 3-year cumulative ORNJ incidence rates were 9.3% and 1.4% in patients with OPC or oral cavity cancer (OCC) and other cancers (*p* < 0.0001), and 2.5% and 8.6% in patients with V60 ≤ 14% and > 14% (*p* < 0.0001). Based on the findings of this first report of the RT dose–volume correlation of the irradiated jaw in a large cohort of HNC patients, the authors suggest using dosimetric parameters, such as V60 ≤ 14% versus >14%, to more accurately estimate the risk of ORNJ for this patient population.

By comparing 68 ORNJ cases to 131 controls, the MD Anderson Head and Neck Cancer Symptom Working Group investigated the dosimetric factors related to ORNJ in OPC patients undergoing IMRT [75]. The maximum doses did not differ statistically, but the mean mandibular dose for the ORNJ cohort was significantly higher (48.1 vs. 43.6 Gy; *p* < 0.0001). All DVH bins in the ORNJ cohort, ranging from V35 to V73, were significantly higher than in the control group (*p* < 0.0006). Two DVH parameters, V44 ≥ 42% and V58 ≥ 25%, were found to correlate with ORNJ rates, and patients with both accounted for 81% of all ORNJ cases.

In a small retrospective study of 36 patients with ORNJ, De Felice et al. [76] found that the Dmean to the affected mandibular bone was 57.6 Gy and that 44% of patients had a D2% ≥ 65 Gy. A multivariate clinical/dose-based Normal Tissue Complication Probability (NTCP) model was developed by van Dijk and colleagues to predict ORNJ_I-IV_ and ORNJ_IV_ after RT or CCRT in patients with HNC [77]. Of the 1259 patients analyzed, 13.7% were diagnosed with ORNJ_I-IV_ and 5.0% with ORNJ_IV_. All of the parameters related to the mandibular dose and volume were found to have a significant correlation with the development of ORNJ in univariate analyses. Both ORNJI_-IV_ and ORNJ_IV_ were found to be independently predicted by D30% and pre-RT dental extraction in multivariate analyses. To reduce the ORNJ_I-IV_ risk to <5%, the model suggested administering a dose of <35 Gy to 30% of the mandible. These findings prove without a reasonable doubt that NTCP models may be used to extrapolate dosage limitations for the mandible, allowing IMRT plans to be optimized for individual patients. This model proposes, for instance, that the mandibular D30% of patients who did not undergo dental extractions before RT should be kept below 42 Gy to achieve a 5% risk of ORNJ development, while the same risk level can only be attainable with a D30% < 35 Gy in patients who had dental extractions. Likewise, D30% should be <17 Gy and <25 Gy in patients with and without dental extractions, respectively, for a more rigid risk threshold of 1%. With respect to ORN_IV_ only, maintaining D30% < 56 Gy without pre-RT dental extractions or D30% < 50 Gy with pre-RT dental extractions may be sufficient to achieve <5% risk of ORN_IV_ development. In the same manner, maintaining D30% < 50 Gy and D30% < 56 Gy with and without pre-RT dental extractions may be sufficient to reach < 5% chance of ORN_IV_ genesis.

As previously discussed, a large-scale study by Aarup-Kristensen and colleagues [8], which included 1224 HNC patients, examined the effect of mandibular dose-volume effects on the incidence of ORNJ following a total dose of 66–68 Gy RT, mostly IMRT. A total of 56 patients (4.60%) were found to have ORNJ. Dosimetric data was available for all 56 cases and 112 controls. For DVH doses between 30 Gy and 60 Gy, significant dose-volume variations were seen between the ORNJ and non-ORNJ groups. Although a direct dose-volume impact was not revealed in this investigation, multivariate analysis revealed a significant correlation between mandibular Dmean and ORNJ rates (HR = 1.04). Cases with ORNJ showed substantially higher Dmean values than controls (41.7 Gy vs. 37 Gy; *p* = 0.02). These data prompted the authors to hypothesize that Dmean represents a general destructive process in a parallel organ, such as the mandible, and that Dmean is an appropriate metric to incorporate in dose planning without a threshold. As a result, the authors recommended keeping the mandibular dosages as low as possible to lessen the likelihood of ORNJ.

Tsai and coworkers examined the records of 402 T1-2 OPC patients who had definitive RT to determine whether there was a correlation between the radiation doses to the mandible and the incidence of ORNJ [78]. In order to do this, a nested case-control study was conducted by matching each ORNJ patient with 1–2 ORNJ-negative individuals. The relative volumes of the mandible exposed to doses ranging from 10 Gy to 60 Gy in increments of 10 Gy were compared using multivariate logistic regression analysis. ORNJ developed in 30 individuals (7.5%). In the matched case-control study, the mean mandibular doses averaged 41.5 Gy for ORNJ patients and 37.5 Gy for ORNJ-negative patients. A correlation between ORNJ and the mean mandibular dosage was found in the univariate analysis but disappeared in the univariate analysis. The percentages of mandibular volumes exposed to doses of 40 Gy and 60 Gy (V40 and V60) differed significantly between the ORNJ and ORNJ-negative groups. The most notable disparity was seen at the V50 (40.5% versus 30.8%; *p* = 0.004). This difference was most pronounced at V50 (*p* = 0.02) after controlling for potential confounders and dental status (dentate or with extraction). In a similar vein, Caparrotti et al. [11] reported that V50 and V60 were associated with ORNJ in a large study involving 1196 OPC patients treated with IMRT.

Lang et al. [79] compared 45 patients without ORNJ to 44 patients with the condition. Dmean > 45 Gy (HR 2.4; 1.0–5.7), Dmax > 60 Gy (HR 1.3; 1.1–2.8), and planned target volume (PTV) proportion > 40% intersection with the mandible (HR 1.1; 1.0–1.1) were found to be significantly associated with ORNJ incidence among the dosimetric factors.

DeLuke et al. [80] sought to determine whether dose volume characteristics were a more accurate predictor of ORNJ than total dosage prescribed to the tumor. The research comprised a total of 56 patients: 27 with ORNJ and 29 matched controls without ORNJ. The statistical models with the dosage variables V50 Gy (cc) and V65 Gy (cc) were more predictive of the occurrence of ORNJ than the model with the total dose. This finding is particularly important because, depending on the RT modality (proton therapy vs. photon therapy) and treatment approach (IMRT vs. 3D-CRT), the precise reflection of the prescribed total tumor dosage in the mandibular doses may differ significantly.

Lee et al. [12] hypothesized that mandibular dose constraints designed to limit high dose to small volumes could prevent ORNJ while not jeopardizing other OAR. 174 OPC patients treated with Volumetric Modulated Arc Therapy (VMAT) were included. The mandibular dose constraint was amended from the HC (historical constraint) of D 0.1 cc 70 Gy to the MCs (modified constraints) of V44 Gy < 42%, V58 Gy < 25%, and D0.5 cc < 70 Gy. In 87% of instances, achieving V44 Gy and V58 Gy without compromising target coverage or OARs resulted in a non-significant drop in osteoradionecrosis (ORN). Mandible V44 Gy and V58 Gy were substantially linked with ORN across all patients (*p* < 0.01 and *p* = 0.03, respectively). V44 Gy was independently and significantly associated with ORN in the HC group (*p* = 0.04).

Recent research by Yilmaz et al. [68] investigated whether pretreatment Hb levels can predict the risk of ORN in 263 patients receiving CCRT for LA-NPC. After a median of 19 months of follow-up (range: 15–34 months), ORNJ was diagnosed in 23 (8.7%) patients. The results revealed that the HPR ≤ 10.6 group had a significantly higher ORN rate (32.5% vs. 1.5%; *p* < 0.001) than the Hb > 10.6 group. The mandibular V59.8 ≥ 36% Gy, pre-CCRT ≥ 4 tooth extractions, the presence of post-CCRT tooth extractions, and the time of post-CCRT tooth extractions > 8 months were the other factors associated with significantly increased ORN rates (*p* < 0.05 for each).

Mandibular Dmean > 40–50 Gy and V40–V60 are the most accurate dosimetric indicators of ORNJ in HNC patients, according to the existing literature. These relatively large disparities across dosimetric predictors may be attributable to differences in RT techniques and dosage prescription, including total and fractional doses. Accordingly, keeping mandibular dosage metrics as low as possible should be the general goal in the absence of established standards in order to reduce the risk of ORNJ without compromising tumor control rates.

## 3. Challenges and Future Prospects

ORNJ remains a difficult-to-manage late complication of RT that imposes significant physical, social, psychological, and economic problems. Although the rate of ORNJ is falling in tandem with advances in RT planning and delivery systems, the total number of ORNJ patients is rising due to rising rates of HNC diagnosis and longer life expectancy. ORNJ causes pain, halitosis, facial deformity, restricted mouth opening, mucosal or cutaneous fistulas, and pathological fractures, all of which have a detrimental effect on fundamental physical functions like swallowing, speaking, and mastication [81,82,83,84]. Delivery, intensity, and tolerance of oncologic therapy may be adversely affected in ORNJ patients due to anemia, infections, leukocytosis, hypo- or hyperproteinemia, hypercoagulation, and cachexia [82]. Consequently, therapeutic response and clinical outcomes may deteriorate, reducing survival prospects. If these patients survive long enough, the trismus and orofacial numbness that might develop after ORNJ may negatively impact every indicator of their QoL [85]. Uncontrolled and persistently progressing ORNJ may also endanger the lives of these patients, either via septicemia or trismus-related problems in intubation during emergencies [86]. Therefore, ORNJ is a serious late RT complication that must be correctly diagnosed and appropriately managed as soon as it manifests.

By definition, ORNJ is a radiation-induced complication, with multiple molecular pathways contributing to its pathogenesis and progression. Consequently, it is crucial to understand the probable radiobiological processes and risk factors related with ORNJ. However, besides the paucity of reliable data on radiobiological processes, there are many challenges in healthy interpretation of the RT-related risk factors, including the modality and technique of RT, and dosage. In addition to the lack of definitive data on radiobiological processes, there are additional obstacles to the accurate interpretation of RT-related risk factors such as modality, technique, and dose. A typical hurdle is the selection of the total recommended dosage for the tumor rather than the mandibular dose-volume estimations based on DVH. In this particular setting, multiple reports have concluded that the recommended tumor dosage is one of the most accurate predictors of ORNJ. A high tumor dose may not always equate to a high mandibular dosage, but still, the mandibular Dmean, Dmedian, and Vx have often been underrepresented, despite their higher predictive values. This fact is especially valid now, in the age of IMRT and IMPT, where mandibular doses may be drastically decreased in comparison to those achieved by traditional RT methods. While a locally progressed OCC treated with a total dosage of 70 Gy in 35 fractions using conventional 2D-RT or 3D-CRT may result in a mean mandibular dose of 70 Gy or more (hot spots), this dose may be easily reduced to 40 Gy or less by employing IMRT. Even if the overall dosage to the tumor remains the same, it is reasonable to expect a substantially higher probability of ORNJ formation with the conventional 2D-RT or 3D-CRT procedures in this particular scenario. Because of this, it may be erroneous to estimate the ORNJ risk purely based on the total dosage administered to the index primary and lymphatic areas in HNC patients. Therefore, we recommend the usage of other dosimetric parameters related to mandible, such as the Dmean and Vx for more accurate risk estimations [87,88,89,90,91,92].

Using the absolute dose parameters numerically, rather than their BED_2_ (biologically equivalent dose of 2 Gy) matching, is a further significant issue. This common practice disregards the significance of dosage per fraction, which is an additional potent driver of any late RT toxicity, including ORNJ. For an OPC patient, for instance, a mandibular Dmean of 46.9 Gy may be attained either after a postoperative total dose of 54 Gy (27 fractions) or after 70 Gy (35 fractions) in the definitive organ-sparing RT strategy. Despite the fact that the mandibular Dmean is identical in both cases, the BED_2_ values for the postoperative and definitive RT settings are 87.7 Gy_2_ and 78.4 Gy_2_, respectively, hence it is reasonable to predict a higher ORNJ risk in the surgically treated patient. Consequently, it is crucial to publish dosimetric findings using BED_2_ values which will provide a meaningful common language that allows for trustworthy comparisons amongst ORNJ research.

Even though tooth extractions and dental implant placements are also prominent risk factors for ORNJ development, they are often overlooked in studies because researchers apply the same dosimetric or clinical parameters for irradiated HNC patients regardless of the absence or presence status, their quantitative burden, or the timing of the procedures relative to RT. Nevertheless, the existing studies and meta-analyses demonstrate that the timing of tooth extractions (before vs. after RT) may massively affect the risk of ORNJ. According to such literature, patients who have tooth extractions after RT have a considerably increased risk of developing ORNJ compared to individuals who had tooth extractions before RT, providing strong support for this fact. Similarly, it is practically unanimously agreed that there is a greater risk of ORNJ if there is an interval of >6 months between RT and post-RT extractions and intervals of <10–14 days between pre-RT extractions and RT. Furthermore, given that tooth extraction and implant placement are invasive and traumatic procedures, their quantity and the affected mandibular volume may serve as reliable indicators of the degree of mandibular trauma and the risk of ORNJ. The findings of the meta-analysis by Jiang et al. showed that the ORNJ risk associated with pre-RT tooth extractions was 4.16%, confirming pre-RT tooth extractions as a risk factor. [93]. Regrettably, rather than the total number of teeth extracted or the affected volume of the mandible, whether or not teeth were extracted served as the primary endpoint in each of the 11 studies that made up this meta-analysis. In this context, the well-designed study recently published by Yilmaz et al. [62] demonstrated that pre-CCRT ≥ 4 tooth extractions were associated with significantly increased ORNJ rates in 263 locally advanced nasopharyngeal carcinoma patients (15.0% vs. 2.3% for < 4 tooth extractions; *p* = 0.005). Nabil et al. [94] extrapolated these findings to the post-RT timeframe and found that 7% of patients undergoing extractions after RT had developed an ORNJ. They also found that the incidence of ORNJ increased by 2% for each additional tooth that was extracted. Notwithstanding the need for additional research, numerous tooth extractions seem to be associated with a considerably higher risk of ORNJ, regardless of whether they are carried out before or after RT. Therefore, to reduce the likelihood of ORNJ incidence and disability, it may be prudent to suggest preserving any repairable teeth in such patients.

The mandibular radiation tolerance limits and associated dose constraints vary considerably between studies, with no generally accepted values. Emami et al. [95] estimated the TD 5/5 for the mandible to be 60 Gy and the TD 50/5 to be 72 Gy when using conventional fractionation to treat the entire mandible. However, these values could be considered conservative. According to the research conducted by Bedwinek et al. [96], the incidence of spontaneous ORNJ was 0%, 1.8%, and 9% at doses of < 60 Gy, 60–70 Gy, and > 70 Gy, respectively. Patients receiving 66 Gy in 2 Gy per fraction doses were merely constrained to a Dmax of 66 Gy in the Radiation Therapy Oncology Group (RTOG) 1016 trial, with no volumetric guidance made [97]. Patients with nasopharyngeal cancer were limited to a Dmax of 70 Gy and a D1cc < 75 Gy in the RTOG 615 protocol for 70 Gy delivered in 33 fractions [98]. Although part of the same research consortium, the mandibular dosage restrictions have shown inconsistent values in the two RTOG trials, exemplifying the continuing conflict in determining dose constraints for the mandible in HNC patients.

Because the risk of ORNJ is affected by factors like pre-irradiation morbidity and irradiated tissue volume, radiation tolerance of the mandible cannot be assessed as a simple dose-response relationship. The risk of ORNJ may rise exponentially if multiple risk factors are present. Disease sites close to the mandible, a dose ≥80 Gy, and the presence of teeth were all factors in one study that predicted an increased risk of ORNJ. Patients with these characteristics were nearly 18 times more likely to develop ORNJ than patients without these risk factors [99]. Some additional factors may hasten ORNJ development and progression as well. To illustrate the situation, a retrospective analysis of 830 HNC patients showed that the addition of chemotherapy to RT can hasten the onset of ORNJ [100]: the time to onset of ORNJ was significantly shorter in patients receiving CCRT than in their RT-alone counterparts (9 months vs. 14 months; *p* < 0.0001). This data supports the development of innovative nomograms that incorporate patient-, disease-, additional intervention-, biomarker- (e.g., inflammation status and related chemokine levels), and dosimetric characteristics to more accurately predict ORNJ rates following RT or CCRT. In the era of individualized patient care, such nomograms may help related medical fields stratify HNC patients into reliable ORNJ risk groups and take the necessary precautions to lower the predicted risk levels.

## 4. Conclusions

The pathogenesis, initiation time, and progression of ORNJ, a harmful late complication of RT or CCRT, may be influenced by a wide variety of confounding factors. However, the current body of evidence suggests that RT-related dosimetric factors are the most influential of all known risk factors. In consideration of the inconsistencies among the recommendations, it seems reasonable to define the whole mandible as a distinct OAR and to maintain the Dmean, Dmax, and Vx values as low as possible without compromising tumor control rates. This is why it’s preferable to use more cutting-edge RT techniques like IMRT and IMPT instead of traditional 2D or 3D RT techniques. Last but not least, to more accurately predict ORNJ rates after RT or CCRT, it is a must to conduct multidisciplinary, large-scale prospective studies taking into account patient, disease, additional intervention, biomarker (e.g., inflammation status and related chemokine levels), and dosimetric characteristics. The findings of such research might be useful in the design of new nomograms and the formulation of guidelines for risk assessment, prompt diagnosis, and tailored treatment of such individuals.

## Authors Contributions

Conceptualization, E.T., A.K., E.S., B.Y., B.P. and U.S.; Methodology, E.T., E.S. and B.Y.; Software, E.T., A.K., E.S. and U.S.; Validation, E.T., A.K., E.S., B.Y., B.P. and U.S.; Formal Analysis, E.T., E.S., B.Y., B.P. and U.S.; Resources, E.T., E.S., B.Y., B.P. and U.S.; Data Curation, E.T., A.K., E.S., B.Y., B.P. and U.S.; Writing—Original Draft Preparation, E.T., E.S., B.Y., B.P. and U.S.; Writing—Review & Editing, E.T., E.S., B.Y., B.P. and U.S.; Visualization, E.T., E.S., B.Y., B.P. and U.S.; Supervision, E.T. and U.S.; Project Administration, E.T., E.S., B.Y., B.P. and U.S. All authors have read and agreed to the published version of the manuscript.

## Figures and Tables

**Figure 1 jcm-12-03025-f001:**
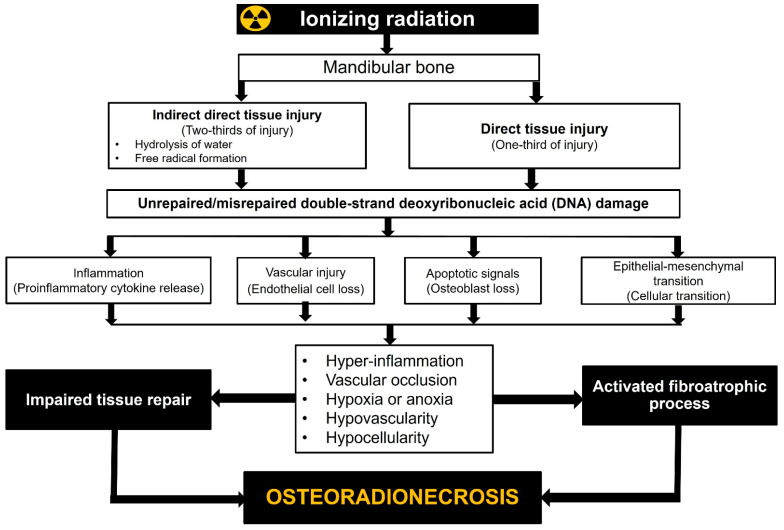
Radiobiological mechanisms of jaw osteoradionecrosis.

**Figure 2 jcm-12-03025-f002:**
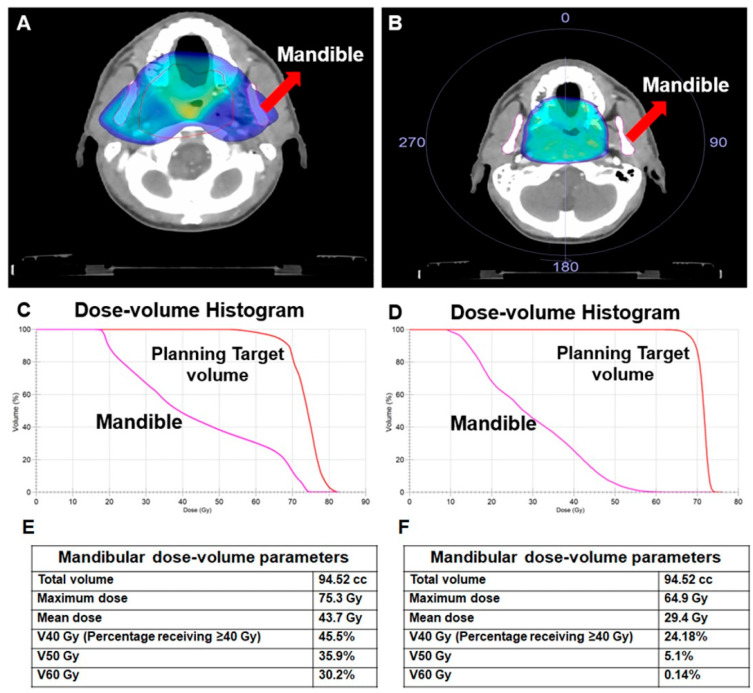
A comparison of two different treatment plans for the same patient with nasopharyngeal carcinoma illustrates the effect of the radiotherapy technique on the doses received by the mandible. (**A**) 3-dimensional conformal radiotherapy (3D-CRT), (**B**) Volume-modulated arc therapy (VMAT), (**C**) Dose-volume histogram for 3D-CRT, (**D**) Dose-volume histogram for VMAT, (**E**) Dosimetric measures for 3D-CRT, (**F**) Dosimetric measures for VMAT.

**Table 1 jcm-12-03025-t001:** Studies investigating the impact of radiotherapy technique and modality on osteoradionecrosis of the jaws rates.

Reference	Study Year	Patients (N)	Tumor Primary	Treatment Technique/Modality	ORNJ Prevalence
Maesschalck et al. [69]	2016	234	HNC	IMRT vs. 3D-CRT	10% vs. 11%
Nguyen et al. [10]	2012	83	OPC	IMRT or IGRT	1.2%
Moon et al. [70]	2017	252	OCC or OPC	IMRT vs. 3D-CRT	5.5%
Aarup-Kristensen et al. [8]	2019	1224	HNC	IMRT	4.6%
Yilmaz et al. [68]	2023	263	NPC	IMRT	8.7%
Balermpas et al. [71]	2022	875	HNC	IMRT	5.0% (all cases with tooth extraction pre-IMRT)
Zhang et al. [72]	2017	584	OPC	IMPT vs. IMRT	2.0% vs. 7.7%
Singh et al. [15]	2022	122	OPC	IMPT	10.6%
Musha et al. [73]	2021	11	HNC	Carbon ion therapy	45.5%

Abbreviations: ORNJ: Osteoradionecrosis of the jaw; IMRT: Intensity-modulated radiotherapy; 3D-CRT: Three-dimensional conformal radiotherapy; HNC: Head and neck cancer; OPC: Oropharyngeal cancer; OCC: Oral cavity cancer; NPC; Nasopharyngeal cancer; IMPT: Intensity-modulated proton therapy; NA: Not available.

**Table 2 jcm-12-03025-t002:** Accessible published dosimetric parameters associated with ORNJ incidence.

Reference	Study Year	Patients (N)	Tumor Primary	Recommended Parameter
Kubota et al. [74]	2021	616	HNC	V60 ≤ 14%
MD Anderson Head and Neck Cancer Symptom Working Group [75]	2017	68	OPC	V44 < 42% V58 < 25%
De Felice et al. [76]	2016	36	HNC	Dmean < 57.6 Gy D2% < 65
van Dijk et al. [77]	2021	1259	HNC	For <5% ORNJ: D30% < 42 Gy (without tooth extraction) For <5% ORNJ: D30% < 35 Gy (without tooth extraction)
Aarup-Kristensen et al. [8]	2021	1224	HNC	Dmean < 37 Gy
Tsai et al. [78]	2013	402	OPC	Dmean < 37.5 Gy V50 (continously) V60 (continously)
Caparrotti et al. [11]	2017	1196	OPC	V50 (continously) V60 (continously)
Lang et al. [79]	2022	89	OCC	Dmean ≤ 45 Gy Dmax ≤ 60 Gy PTV proportion intersecting the mandible ≤ 40%
DeLuke et al. [80]	2022	83	HNC	V50 (continously) V65 (continously)
Lee et al. [81]	2022	174	OPC	V44 (continously) V58 (continously)
Yilmaz et al. [68]	2023	263	NPC	V59.8 ≥ 36% Gy

Abbreviations: HNC: Head and neck cancer; OPC: Oropharyngeal cancer; OCC: Oral cavity cancer; NPC; Nasopharyngeal cancer; ORNJ: Osteoradionecrosis of the jaw; Vx: Volume receiving X Gray or higher dose: Dx: Percentage of the prescription dose received by the X% of the mandible; Dmean: Mean dose: Dmax: Maximum dose; PTV: Planning target volume.

## Data Availability

For researchers who satisfy the criteria for access to sensitive data, the datasets utilized and/or analyzed during the current study are accessible from the Baskent University Department of Radiation Oncology Institutional Data Access: adanabaskent@baskent.edu.tr.

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
