# Peer review of "Review of Osteoradionecrosis of the Jaw: Radiotherapy Modality, Technique, and Dose as Risk Factors"

_jcm, 2023, doi:10.3390/jcm12083025_

Round 1
Reviewer 1 Report
I'd like to thank the author for this extensive literature review for a critical aspect of late radiation induced toxicity, one of the most harmful.
I have no major revision to make.
Some comments:
- in the abstract (line 2) and in the introduction (line 56) you stated "effective but intense", I don't feel appropriate as adjective, I would delete
- introduction (line 45) "only curative option", surgery is a curative option for early stages HNC, should be mentioned since surgery is as effective as radiation in early stages
- introduction (line 48) "neoadjuvant", radiation is never given as neoadiuvant therapy in HNC
- introduction (lines 64-65) I would say "Although the impact of improved dental care programs cannot be 64 underestimated, this reduction in ORNJ incidence is primarily due to the highly conformal dose
- line 213 typo -SMA
- line 221 typo TGF--B1
Author Response
Reviewer 1
We would like to thank the reviewers for their thoughtful and helpful remarks, which we feel will considerably enhance the quality and consistency of our current work. We made every effort to perform the necessary revisions as addressed individually.
Comment 1- in the abstract (line 2) and in the introduction (line 56) you stated "effective but intense", I don't feel appropriate as adjective, I would delete
Response 1- The "effective but intense" phrase has been replaced by "aggressive" in both sentences.
Comment 2- introduction (line 45) "only curative option", surgery is a curative option for early stages HNC, should be mentioned since surgery is as effective as radiation in early stages
Response 2- The reviewer is correct. However, we meant medically inoperable patients and have amended the corresponding sentences.
Comment 3- introduction (line 48) "neoadjuvant", radiation is never given as neoadiuvant therapy in HNC
Response 3- We appreciate the reviewer's thoughtful reminder. According to the recommendation, "neoadjuvant" has been removed from the revised text.
Comment 4- introduction (lines 64-65) I would say "Although the impact of improved dental care programs cannot be 64 underestimated, this reduction in ORNJ incidence is primarily due to the highly conformal dose
Response 4- We appreciate the thoughtful comment. The corresponding sentence in the updated text has been reworded to better reflect the suggestions made.
Comment 5- line 213 typo -SMA
Response 5- SMA has been removed from the text to fix this editing mistake.
Comment 6- line 221 typo TGF--B1
Response 6- Typo has been corrected.
Reviewer 2 Report
The authors well-summarize the mechanisms by which osteoradionecrosis of the jaw occurs following radiotherapy and the risk factors in this review. It is so valuable that this review includes biological evidence regarding osteoradionecrosis as well as clinical insights. This topic does not sound new but should provide beneficial information for researchers and clinicians. please check the following points.
1. In some parts, two hyphens are put between TGF and β1 (TGF-- β1).
2. Figure 1: The authors describe "1/3" in the boxes of injury. What does it mean?
3. Table 2: Reference numbers are not put next to the author's name for some references. Because the number is put for the other references in Tables 1 and 2, the authors should do as well.
Author Response
We would like to thank the reviewers for their thoughtful and helpful remarks, which we feel will considerably enhance the quality and consistency of our current work. We made every effort to perform the necessary revisions as addressed individually.
Comment 1. In some parts, two hyphens are put between TGF and β1 (TGF-- β1).
Response 1. Such editing errors have been identified and addressed.
Comment 2. Figure 1: The authors describe "1/3" in the boxes of injury. What does it mean?
Response 2. They represent the extent of tissue damage produced by ionizing radiation. Corrections have been made as needed.
Comment 3. Table 2: Reference numbers are not put next to the author's name for some references. Because the number is put for the other references in Tables 1 and 2, the authors should do as well.
Response 3. As indicated, necessary adjustments have been made throughout the entire manuscript.
Reviewer 3 Report
Title
I suggest changing the title to: Review of Osteoradionecrosis of the Jaw: Radiotherapy Modality, Technique, and Dose as Risk Factors
Introduction
- Provide references/citations for lines 44-51
- Lines 63-64: “incidence of ORNJ is currently less than 5–6% in the majority of publications” but the authors cite only two references [5,6]
- What is the difference between RT modality and RT technique? I can see the answer coming up later in the manuscript but should be explained earlier.
Lines 119-123: can you explain why faster devitalization occurs in the mandible although it has higher turnover rate?
Page 9 lines 336-346: the study by Maesschalck (reasonable sample size) showed no difference in ORNJ prevalence but the P value was not significant. The study by Nguyen et al (small sample size compared to Maesschalck) you need to provide the reader with the p value.
Page 9 lines 351-355: you provide the HR and p value to all variables for Moon et al but not for the variable 3D-CRT vs IMRT?
Lines 496-497: I think you mean multivariate analysis at the end of the sentence.
Figure 1: what is EMT?
Table 1: what does the P in the last column (P= 1.0 and P= NA) stand for?
Table 2: Didn’t De Felice et al report the primary tumor?
Author Response
We would like to thank Reviewer 3 for his/her insightful comments. The following is our reply to each comment:
Comment 1. I suggest changing the title to: Review of Osteoradionecrosis of the Jaw: Radiotherapy Modality, Technique, and Dose as Risk Factors
Response 1. We changed the title as recommended.
Comment 2. Provide references/citations for lines 44-51
Response 2. The recommended addition of references has been made to each sentence.
Comment 3. Lines 63-64: “incidence of ORNJ is currently less than 5–6% in the majority of publications” but the authors cite only two references [5,6]
Response 3. Additional pertinent references have been added.
Comment 4. What is the difference between RT modality and RT technique? I can see the answer coming up later in the manuscript but should be explained earlier.
Response 4. The difference between the two terms have now been described in the text (Lines 80-84) as recommended.
Comment 5. Lines 119-123: can you explain why faster devitalization occurs in the mandible although it has higher turnover rate?
Response 5. The matter in question has been appropriately dealt with in the revised document as required.
Comment 6. Page 9 lines 336-346: the study by Maesschalck (reasonable sample size) showed no difference in ORNJ prevalence but the P value was not significant. The study by Nguyen et al (small sample size compared to Maesschalck) you need to provide the reader with the p value.
Response 6. Since all patients received IMRT or IGRT, Nguyen's study was observational and not a direct comparison between treatment techniques. Therefore, there was no P-value provided.
Comment 7. Page 9 lines 351-355: you provide the HR and p value to all variables for Moon et al but not for the variable 3D-CRT vs IMRT?
Response 7. The revised text now includes HR and P-value.
Comment 8. Lines 496-497: I think you mean multivariate analysis at the end of the sentence.
Response 8. Yes, we meant multivariate analysis, and the revised manuscript has been modified accordingly.
Comment 9. Figure 1: what is EMT?
Response 9. EMT refers to epithelial-mesenchymal transition. The modifications have been made to Figure 1 as per the requirements.
Comment 10. Table 1: what does the P in the last column (P= 1.0 and P= NA) stand for?
Response 10. They were P-values. However, we removed them from the table for consistency of the Table as a whole.
Comment 11. Table 2: Didn’t De Felice et al report the primary tumor?
Response 11. They were head and neck cancers. This editing issue has now been resolved.
Reviewer 4 Report
Dear authors,
This is a comprehensive review of osteoradionecrosis of the jaw.
In their review, the biological background in its development and its relevance to the irradiation dose and treatment techniques used are mentioned in detail. The content is interesting and based on in-depth bibliographical research and should be of interest to readers.
However, I believe that several minor corrections which follow would improve this review remarkably.
1. p1, l45: The phrase "because it is the only curative option for..." is not correct. RT is used not only to treat early disease or salivary gland cancer but also to treat locally advanced disease. Please consider generalizing the role of RT in the treatment of HN cancer.
2. p8, Figure 2: The CT images illustrate the dose reduction to the mandibular ramus with VMAT. However, I think that images of the slices which contain the alveolar part (or the mandibular body, as mentioned later in the article) of the mandible is more appropriate in this occasion because the dose to it is regarded to associate with the occurrence of ORN.
3. p12, l498: The phrase "disappeared in the univariate analysis" should be "in the multivariate analysis" instead ?
Please consider corrections according to the comment above.
Regards,
Author Response
We couldn't upload the revised manuscript as there was the button available for this function, therefore we uploaded the revised manuscript as an attachment to Reviewer 4 please make available the system for our upload or upload the revised manuscript on behalf of us

Round 2
Reviewer 3 Report
Thank you for addressing the comments.